# Short-term prediction of COPD exacerbations based on wearable vital sign monitoring

Florian Tilquin[1], Sylvain Le Liepvre [1]*, Soumaya Balbolia [1], Marie Pirotais[1], Yann Le Guillou[1], Jean-Claude Cornu[2], Nicolas Roche[3,4], Gerard Criner[5], Marie Joyeux-Faure[6], Jean-Louis Pépin[6]

1 Biosency, Cesson Sevigné, France, 2 Verdun Hospital, Verdun, France, 3 Department of Pulmonology, Hôpital Cochin, AP-HP, APHP-Centre, Paris, France, 4 Université Paris Cité, Institut Cochin, INSERM U1016, Paris, France, 5 Department of Thoracic Medicine and Surgery, Lewis Katz School of Medicine at Temple University, Philadelphia, Pennsylvania, United States of America, 6 HP2 Laboratory, INSERM U1300, Grenoble Alpes University, Grenoble, France

* sylvain.le.liepvre@biosency.com

## Abstract

Early detection of acute exacerbations of chronic obstructive pulmonary disease (AECOPD) remains a critical challenge in COPD management. This study introduces the Bora Vital Sign Standard Score (BVS³), a novel unsupervised statistical score that predicts AECOPD using vital signs collected at home through remote patient monitoring, and retrospectively evaluates its predictive performance in identifying AECOPD ahead of clinician-defined episodes. The eMEUSE-SANTÉ clinical trial (NCT04963192) involved 220 COPD patients who were remotely monitored for six months using a CE-certified (Class IIa) connected wristband measuring oxygen saturation ($SpO_2$), breathing rate (BR) and heart rate (HR). A total of 42 physician validated exacerbations of COPD with no missing remote monitoring data were documented in 39 patients at a general hospital. Continuous 24-hour monitoring of vital signs using a connected wristband was well accepted over the long term, with a median adherence of 86% indicating strong patient compliance. The BVS³ risk score achieved excellent predictive performance, with an AUC of 0.88 (95% CI 0.83 – 0.92) for moderate and severe AECOPD combined. The BVS³ score anticipated exacerbations an average of $4.4 \pm 3.1$ days before clinical confirmation, with an overall accuracy of 84.8% and sensitivity of 74% with 85% specificity. Individual Z-scores for heart rate (z-HR), breathing rate (z-BR) and oxygen saturation (z-$SpO_2$) showed specific predictive capabilities for moderate and severe events, yielding AUCs of 0.83, 0.82 and 0.71 respectively, but with inferior performances compared with the combination of the 3 vital signs Z-scores. These results demonstrate that integrating passive remote monitoring with unsupervised statistical modeling provides a scalable, high-compliance approach to AECOPD detection. The interpretable BVS³ risk score achieves good accuracy and anticipation for AECOPD prediction with minimal

**Data availability statement:** The datasets generated and analyzed during this study are not publicly available due to the risk of re-identification of individual participants, in compliance with the General Data Protection Regulation (GDPR). The dataset is securely maintained in the MARS academic repository, managed by the HP2 laboratory, Université Grenoble Alpes. (https://hp2.univ-grenoble-alpes.fr/en) Access to the data may be granted for legitimate research purposes, subject to approval by the HP2 laboratory and appropriate data-sharing agreements. Requests for access should be directed to the institutional contact point: contact.mars@chu-grenoble.fr.

**Funding:** This work was supported by the French National Research Agency (ANR) in the framework of the "Investissements d'avenir" program "Territoires d'Innovation» and by "La banque des territoires (Groupe Caisse des dépôts)" (eMEUSE-SANTÉ program to FT, SL, SB, MP, YG, JC, MJ, and JP). JP and MJ are also supported by the French National Research Agency (ANR) in the framework of the "FRANCE 2030" program, the "e-health and integrated care" chair of Grenoble Alpes University Foundation, and the "Sleep Health-AI chair" in the "MIAI Cluster" of artificial intelligence (ANR-23-IACL-0006 to JP and MJ). JP reports income related to medical education from RESMED, SEFAM, Zoll-Respicardia, Eli Lilly, Idorsia, Pharmanovia, Biosency, and Bioprojet. NR reports research funds and fees from Chiesi, Pfizer, and GSK, and personal fees (advisory boards, consultancy, education, and presentations) from Austral, Biosency, MSD, AstraZeneca, Chiesi, Menarini, Nuvaira, Roche, Sanofi, and Zambon. GC reports research grants from AstraZeneca, Boehringer Ingelheim, Broncus, Chiesi Farmaceutici, Corvus, Fisher-Paykel Healthcare, Genentech, Gilead, GlaxoSmithKline, Lilly, MedImmune, National Institute of Health – National Heart, Lung, and Blood Institute, Novartis, Olympus, PA-DOH, Pearl, Pfizer, PneumRx, Pulmonx, Regeneron, Respironics, Roche, and Spiration; and consultation fees from Almirall, AstraZeneca, Broncus, BTG, CSA Medical, EOLO, GlaxoSmithKline, Mereo, Nuvaira, PneumRx, and Pulmonx. The funders had no role in study design, data collection and

patient burden. By enabling earlier intervention, this end-to-end digital solution could significantly improve patient outcomes through proactive disease management.

## Author summary

Chronic obstructive pulmonary disease (COPD) is a long-term lung condition that can suddenly worsen, causing irreversible lung function decline, hospitalization or even death. Predicting these acute exacerbations of COPD in everyday care remains a major challenge for patients and healthcare teams, largely because existing digital tools that perform well in research settings often prove impractical in real life. Although advanced multi-sensor systems such as those reported by Wu et al. and Atzeni et al. have reported high prediction performance their cost, setup complexity, and patient burden limit widespread adoption. In our study, we explored whether a single, CE-marked, medical-grade connected wristband that passively records heart rate, breathing rate, and blood oxygen saturation level combined with a novel unsupervised, statistical model-based risk score, could offer a practical and effective solution for predicting exacerbations of COPD in routine care. Our goal was to achieve high predictive performance while minimizing patient involvement to ensure real-world feasibility. We developed a fully interpretable risk score, called BVS[3], using statistical modeling to detect early signs of exacerbation from these three vital signs. We evaluated both the score's predictive performance and patient adherence using data from more than two hundred patients monitored for six months. The BVS[3] score demonstrated excellent predictive performance, detecting exacerbation events an average of 4.4 ± 3.1 days before onset with an overall accuracy of 84.8% (AUC 0.88), while patients maintained high adherence to the wristband (86%), suggesting its potential role as an early warning tool to support early clinical review and management decisions within remote patient monitoring programs. By combining continuous home monitoring with artificial intelligence, this work shows a practical way to support earlier intervention, reduce hospitalizations, and improve daily management for people living with COPD.

## Introduction

Chronic Obstructive Pulmonary Disease (COPD) is one of the most burdensome chronic respiratory disease owing to associated morbidity, mortality and health related costs [1–3]. The disease burden put on the health system is essentially related to the disability resulting from the disease, comorbidities and acute exacerbations of COPD (AECOPD) [4].

AECOPD are defined as events characterized by dyspnea and/or cough and sputum that worsen over ≤14 days [5]. A worsening of respiratory symptoms may require additional treatment and can lead to hospitalization for the most severe

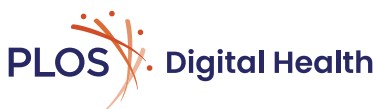

analysis, decision to publish, or preparation of the manuscript.

**Competing interests:** YG and MP are co-founders and employees of Biosency. FT and SL are employees of Biosency and are named inventors on patents WO2024153532A1 and EP4403096A1 issued to Biosency. SB is an employee of Biosency. JP, NR, and GC serve as members of Biosency's scientific advisory board. All other authors have declared that no competing interests exist.

**Abbreviations:** AUC, Area under the curve; AECOPD, Acute exacerbations of chronic obstructive pulmonary disease; BVS³, Bora vital sign standard score; BR, Breathing rate; COPD, Chronic obstructive pulmonary disease; CRP, C-reactive protein; ePROMs, Electronic patient-reported outcome measures; GOLD, Global Initiative for Chronic Obstructive Lung Disease; HCPs, Healthcare professionals; HR, Heart rate; IQR, Interquartile Range; NIV, Noninvasive ventilation; OSA, Obstructive sleep apnea; PPG, Photoplethysmography; QoL, Quality of life; ROC, Receiver operating characteristic; RPM, Remote patient monitoring; $SpO_2$, Oxygen saturation; Z-BR, Breathing rate z-score; Z-HR, Heart rate z-score; $Z-SpO_2$, Oxygen saturation z-score.

exacerbations. AECOPD are common, affecting up to 70% of COPD patients [6]. AECOPD are associated with substantial impairment in quality of life [7], irreversible lung function decline, increased morbidity, mortality [8] and healthcare related costs [9]. Studies have shown that a significant proportion of exacerbations are unreported although they affect quality of life and may accelerate disease progression [10]. Additionally, untreated exacerbations are associated with an increased risk of hospitalization, and delayed initiation of treatment is associated with longer treatment duration [11]. Optimizing AECOPD early detection and management is therefore both a clinical and medico-economic priority in COPD care.

A promising approach relies on the development of prediction/detection tools available for healthcare professionals (HCPs) to enable early detection of AECOPD and initiation of timely management. This early intervention will facilitate diagnostic confirmation, identification of etiological causes (e.g., viral testing, sputum culture), and the implementation of a personalized care plan. By intervening early, HCPs can significantly reduce symptoms and the severity of AECOPD episodes, thereby minimizing their impact on patients' health status, hospitalization rates and readmission rates, and overall disease progression [6, 12].

The risk of AECOPD can be predicted by combining patient-specific risk factors such as smoking status, prior exacerbations, comorbidities and clinical phenotype with established AECOPD environmental triggers (i.e., viruses, pollution) and symptoms, airflow limitation, or biomarkers [13, 14–16].

Remote home patient monitoring (RPM) has gained a growing interest as a relevant approach for anticipating AECOPD. Recent advances in wearable technologies have enabled continuous, non-invasive monitoring of a wide range of biomarkers, including vital signs, physical activity, sleep, and cough-related signals, in real-world settings [17–21]. Wearable platforms such as wrist-worn devices, epidermal patches, and smart rings have demonstrated utility for the early detection of clinical deterioration across diverse clinical pathways, including perioperative surveillance, infectious diseases, and heart failure [22, 23]. For predicting AECOPD, RPM approaches have progressed from modestly accurate digital questionnaires to sophisticated multi-sensor systems with higher predictive capabilities [13,24,25–32]. The effectiveness of this approach is hindered by patient-level barriers, including low digital literacy and engagement, and by healthcare provider-level barriers such as increased workload and limited resources or funding [33]. These issues must be addressed for successful implementation in routine care. Recently, the Rome proposal [5] introduced a standardized framework for grading AECOPD severity based on measurable variables including dyspnea, heart rate (HR), breathing rate (BR), oxygen saturation ($SpO_2$), arterial carbon dioxide pressure ($PaCO_2$), and C-reactive protein (CRP). Among these, HR, BR, and $SpO_2$ are particularly suited to real time detection of AECOPD because they can be measured continuously and non-invasively with commercially available wearable devices.

In this context, we previously described a methodological framework for risk estimation in a patent [34]. In the present work, we introduce the Bora Vital Sign Standard Score (BVS³), a specific instantiation of this risk score framework derived using

unsupervised, statistical techniques applied to vital signs data (heart rate, breathing rate, $SpO_2$) collected through RPM via a connected wristband. By focusing on HR, BR, and $SpO_2$, the score is fully interpretable for clinicians, its primary users, while also maximizing scalability, simplicity, and patient adherence, relying solely on continuously and non-invasively measured vital signs from a single device.

The $BVS^3$ score incorporates two key innovations, detailed in the Methods section, that enhance AECOPD prediction: a temporal filter to reduce false alerts arising from short-term variations in vital signs, and a Gaussian Process filter to manage randomly missing data. This score correlates with acute deteriorations of the patient's health status [35, 36], suggesting that in routine care it could support the early detection of AECOPD by triggering alerts to healthcare professionals when a predefined threshold is crossed.

From a clinical perspective, automated AECOPD detection tools must align with healthcare professionals' expectations and operational constraints. Clinicians typically prioritize the early identification of moderate to severe exacerbations, which represent clinically actionable episodes requiring therapeutic interventions such as systemic corticosteroids, antibiotics, or modification of oxygen therapy, in accordance with current GOLD recommendations [37]. Early identification of such events may help reduce symptom severity, hospitalization and readmission rates, and overall disease progression.

Because AECOPD occur relatively infrequently (approximately one to two events per patient per year on average), the positive predictive value of a daily alerting system is intrinsically limited. Under these conditions, even models with high sensitivity generate a substantial number of false-positive alerts, as prediction are evaluated predominantly on days without an exacerbation. Consequently, the clinical utility of an AECOPD short-term prediction system depends primarily on constraining the alert burden to a clinically acceptable level for healthcare professionals, and subsequently maximizing sensitivity within this predefined alert rate. Previous studies of remote patient monitoring systems suggest that an alert frequency of approximately five to ten alerts per patient per year is acceptable for healthcare professionals [38, 39], whereas higher alert rates have been associated with reduced adoption [40].

In the present study, we therefore defined a maximum acceptable alert burden of approximately six alerts per patient per year as the target operating point for short-term AECOPD prediction.

The objective of this retrospective study was to evaluate the clinical performance of the $BVS^3$ score in predicting physician validated moderate and severe AECOPD events. Specifically, we assessed the precision and anticipation with which $BVS^3$ could identify such events within the 10 days preceding onset, with the aim of supporting clinician decision making through early alerts while limiting unnecessary false alarms. We hypothesized that $BVS^3$ would provide clinically meaningful early-warning capability, supporting its potential role in real-world COPD management.

## Materials and methods

### Study design

The present work is a retrospective ancillary analysis of the eMEUSE-SANTÉ trial (NCT04963192), a monocentric prospective observational study conducted in France from 2021 to 2024, evaluating a home-based, multidisciplinary-integrated care in chronic respiratory diseases including COPD and Obstructive Sleep Apnea (OSA), living in rural areas of a French county with limited access to conventional healthcare. This integrated care approach incorporated a CE-marked connected wristband, enabling remote monitoring of heart rate, breathing rate and $SpO_2$ over a six-month-follow-up period for COPD patients.

For this analysis, only patients with COPD were included. Importantly, while data collection was prospective, the $BVS^3$ score was not used in patient management: nurses monitored raw vital signs without algorithm-derived alerts or intervention. For the present analysis, the $BVS^3$ algorithm was retrospectively applied to the prospectively collected dataset to validate its predictive performance against physician validated AECOPD events.

The rationale for using this dataset was its real-world representativeness, as it enrolled a broad COPD population without restrictive selection criteria, reflecting routine clinical practice in a rural healthcare context.

Crucially, the BVS[3] score is an unsupervised statistical model algorithm that applies predefined mathematical formulas to physiological time-series data, using Gaussian processes to model and process the signals. It does not require training on AECOPD outcomes, nor any parameter fitting. Consequently, the present work constitutes an external validation of the score, the dataset served exclusively as a validation set.

### Inclusion criteria

Eligible participants were all patients with COPD attending a general hospital respiratory medicine consultation (Verdun, Meuse county, France), with or without noninvasive ventilation (NIV) and/or oxygen therapy, without restriction on the number of participants in each group. Participants were required to be able to use a smartphone and be affiliated with the French national social security health insurance scheme.

### Ethics

The eMEUSE-SANTÉ study was approved by the ethics committee CPP Sud-Est III on June 1, 2021 (No. 21.04.16.39948). Written informed consent was obtained from all participants before inclusion in the study.

### Data collection

Participant enrollment in the eMEUSE-SANTÉ study began in May 2021 and the last follow-up visit was completed in August 2024. The eMEUSE-SANTÉ study flow is presented in Fig 1. At inclusion, patients were equipped at home with a connected wristband (Bora band® model BB100, Biosency), a CE-certified Class IIa medical device compliant with ISO 80601-2-61:2017 standard, providing CE-marked measurements of heart rate (HR), breathing rate (BR), and oxygen saturation ($SpO_2$) based on photoplethysmography (PPG) signals. The wristband was automatically measuring HR, BR, $SpO_2$, and physical activity (step count and activity duration) every 10 minutes. Data were transmitted to a CE-marked class IIa remote monitoring platform (Bora connect®, Biosency) via a gateway (Bora Box, Biosency).

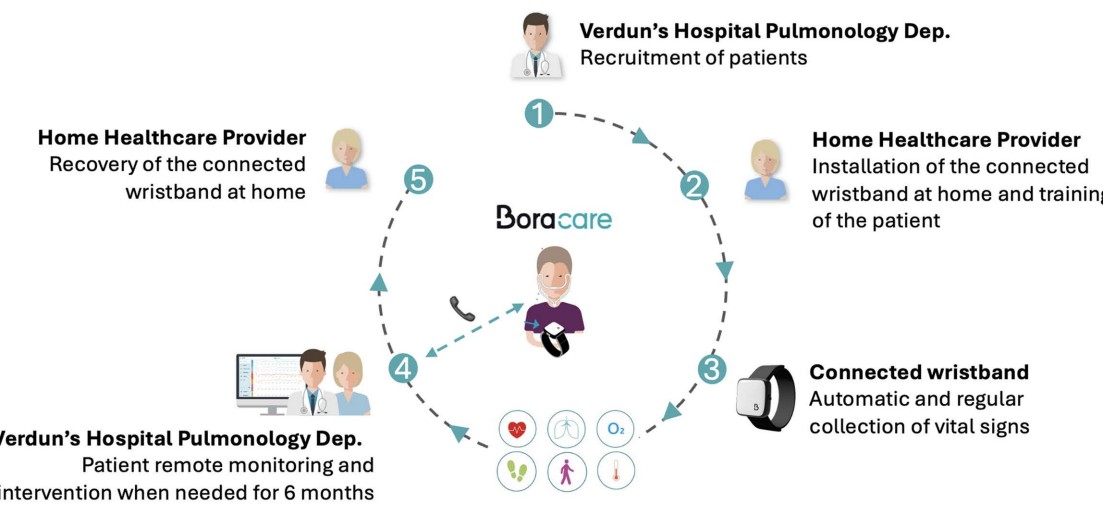

**Fig 1. Description of the eMEUSE-SANTÉ study flow.**

Patients were instructed to wear the monitoring wristband continuously around the clock, except during bathing. A remote patient monitoring (RPM) session was defined as a continuous period during which a patient was equipped with the connected wristband, the maximum duration being the planned monitoring duration of 6 months. The Pulmonology Department of Verdun's General Hospital conducted the medical follow-up with case managers (nurses) respiratory physicians objectively documenting exacerbations. Follow-up interviews were carried out every three months and were used to identify additional moderate exacerbations managed by general practitioners.

According to current international guidelines [37], moderate exacerbations were defined as those treated with oral corticosteroids and/or antibiotics, while severe exacerbations were defined as events necessitating hospitalization or an emergency department visit. The dates of analyzed exacerbations were defined as the day on which the medical consultation and corresponding prescription occurred, or the day of emergency department visit or hospital admission, and were required to coincide with an active RPM session.

## Statistical analysis and estimation of the score performance

Patient characteristics were reported using means with standard deviations (SD) or medians with interquartile ranges (IQR), as appropriate based on variable distribution.

No *a priori* sample size calculation was performed. Statistical significance of the score's predictive performance was subsequently assessed using appropriate tests (t-tests and confidence intervals for performance metrics), ensuring that the observed associations were unlikely to have occurred by chance. Bootstrap resampling was used to estimate 95% confidence intervals for predictive performance metrics, as the score was predefined and not trained on outcome data.

To evaluate the performance of the BVS³ score, we analyzed non-overlapping 10-day periods of collected data. The 10-day period was chosen based on clinical reasoning and prior literature on AECOPD prediction [13, 15, 26, 28, 41] which has typically used 7 days or 14 days periods. In our pilot study [36], detections occurred as early as 8 days before an AECOPD event, suggesting that a 7-day period was insufficient. Conversely, a 14-day period was considered too long from a clinical perspective, since alerts occurring this far in advance could be dissociated from the actual AECOPD, given that 14 days corresponds to the upper limit of AECOPD development [5]. Therefore, the 10-day period was chosen as a compromise, balancing early detection with clinical relevance.

The BVS³ score is intended to be used as a longitudinal risk monitoring tool in patients with established COPD during RPM sessions. The clinically relevant question is how often the score generates alerts during periods in which no AECOPD is clinically observed, which corresponds to the false alert burden under realistic deployment conditions, and how often the score generates alerts during periods in which an AECOPD is observed to estimate the number of AECOPD that would have been predicted.

Three types of time periods were defined relative to the AECOPD event (Fig 2): the case, control, and dismissed periods. The case period was defined as the 10-day period immediately preceding an AECOPD validated by clinicians (D-10 to D0). The control period corresponded to 10-day periods in non-exacerbating patients, or to 10-day periods beginning at least 30 days before (D-30) or 30 days after (D+30) an AECOPD in exacerbating patients. As illustrated in Fig 2, periods from D-20 to D-10 and from D0 to D+30 were excluded from control periods to minimize contamination by pre-exacerbation signals and post-exacerbation recovery effects, which may persist for several weeks.

True positives were defined as the presence of a detection by the risk score within a case period, while false positives were defined as the presence of a detection by the risk score within a control period. The dismissed period, defined as the time period between 20 and 10 days prior to an exacerbation event, was excluded from the computation of the score performance as detections in this time period could not be reliably attributed to the exacerbation event or to an unrelated physiological change.

The receiver operating characteristic (ROC) curve was computed based on the number of true and false positives generated by different alert thresholds on the evaluated scores. The area under the curve (AUC) of the ROC was the primary

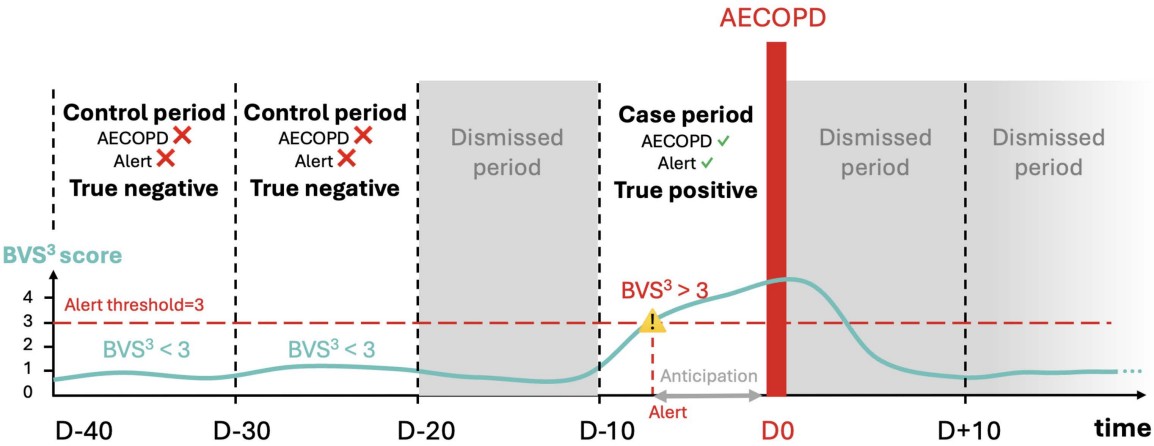

**Fig 2. Definition of case and control periods for the estimation of the risk score performance in predicting AECOPD.**

outcome used to evaluate the score performance. Accuracy and sensitivity were calculated at a specificity level of 85% to illustrate differences in performance between combinations of variables. Confidence intervals for the AUC and other performance metrics were estimated using a stratified, non-parametric percentile bootstrap with 1000 resampling iterations.

The ability of the BVS³ score to discriminate between case and control periods was evaluated across the full range of alert thresholds using receiver operating characteristic (ROC) analysis.

In accordance with the predefined clinically acceptable alert burden described in the Introduction, we report the performance of the BVS³ score at a fixed false-positive rate of 15%, corresponding to approximately six alerts per patient per year when using a 10-day prediction horizon. The alert threshold was defined as the BVS³ score value yielding this target specificity on the ROC curve. Because threshold selection was based solely on a fixed specificity criterion and did not involve outcome-based optimization, this approach minimizes overfitting concerns associated with data-driven threshold calibration. The resulting numeric threshold (BVS³ = 3) serves only to implement this predefined operating point and alternative scalings of the score would yield equivalent relative performance.

## BVS³ score computation

The selection of HR, BR and $SpO_2$ was guided by pathophysiological relevance, as described in the Rome proposal [5]. In that work, a panel of experts systematically evaluated 21 candidate variables identified through a comprehensive literature review within a conceptual framework describing the causes, pathobiological mechanisms, and physiological consequences of AECOPD. An AECOPD severity assessment scheme for both clinical evaluation of patients and research and clinical trials was agreed upon through expert consensus, based on five easy-to-evaluate parameters (dyspnea, HR, BR, $SpO_2$, CRP) for mild and moderate AECOPD, with the addition of arterial blood gas analysis ($PaCO_2$ and pH) required to classify severe AECOPD. Within this expert-ranked set of measurable variables, HR, BR, and $SpO_2$ were selected for the development of the BVS³ score because they can be measured continuously and non-invasively with commercially available wearable devices, making them particularly suitable for real-time detection of AECOPD. By contrast, dyspnea assessment requires patient-reported input such as a visual analogue scale [5], limiting patient acceptability over the long-term; CRP measurement requires blood sampling, which is invasive and costly [42], and $PaCO_2$ measurement relies on arterial blood gas analysis, which is invasive and impractical for home monitoring [43].

Conceptually, the BVS³ score quantifies how much a patient's heart rate (HR), breathing rate (BR), and oxygen saturation ($SpO_2$) deviate from their usual baseline. Each vital sign is continuously compared with the patient's historical average

using a z-score, which expresses deviation from the two-week mean in standard deviation units, creating a personalized reference. The BVS[3] score increases when HR and BR rise and SpO$_2$ falls relative to baseline, as typically occurs in the early phase of an exacerbation. When the score exceeds a predefined threshold, it indicates that the patient's physiological status has changed in a clinically meaningful way that may precede an AECOPD event.

The following computational steps were applied to implement this concept. To ensure robust analysis of vital signs collected by the Bora band®, vital sign time series were resampled from 10-minute to hourly intervals using a median filter to mitigate possible noise in the vital sign measurements, arising both from the photoplethysmography (PPG) signal processing algorithm and from natural noise inherent to the patient's activity. Outliers were identified and filtered separately for each vital sign time series using a threshold of four robust standard deviations, estimated via the median absolute deviation (MAD). Gaussian process filtering [44,45] with a Matérn kernel was applied using a 30-day sliding window, with hyperparameters optimized by maximizing the log-likelihood, to denoise the vital sign time series and interpolate missing values and capture underlying trends in a flexible, probabilistic framework. The resulting data were evenly sampled at one-hour intervals with no missing values. To reduce noise, we then apply a low-pass filter to each time series.

For each processed vital sign time series, a Z-score time series was computed by comparing current values to the baseline values, derived from the average and the standard deviation of the patient's past vital sign measurements. This approach allowed deviations to be assessed in a personalized, patient-specific manner. The three vital sign Z-scores time series were aggregated into a single composite time series using the survival function of the normal distribution, with each contributing equally to capture the simultaneous occurrence of relative anomalies across vital signs, as described by the equations (1) and (2). Briefly, at each time point t, the Z-scores of heart rate (Z-HR), respiratory rate (Z-BR), and oxygen saturation (Z-SpO$_2$) are first averaged. This average is then scaled and passed through the survival function, which returns the probability of observing a deviation as large or larger under a standard normal distribution. Taking the negative logarithm of this probability produces the BVS[3] [34], such that time points with simultaneous large deviations in multiple vital signs yield higher scores.

$$BVS^3(t) = -\log\left( sf\left[ 2\left( \frac{Z_{HR}(t) + Z_{BR}(t) + Z_{SpO_2}(t)}{3} \right) \right] \right)$$

(1)

$$sf(x) = \frac{1}{\sqrt{2\pi}} \int_x^\infty e^{-t^2/2}\, dt$$

(2)

More detailed mathematical derivations and implementation steps for the computation of the BVS[3] score are provided in S1 Text.

Fig 3 illustrates the computation of the risk score, showing vital sign time series (markers), Gaussian Process filtering (solid lines), the BVS[3] risk score (black line) and its associated alert threshold corresponding to a false positive rate of 15%. This illustrative case depicts data from a patient exhibiting a moderate exacerbation. During the initial stable period (D0-30 to D0-10), vital signs show non-significant or non-simultaneous fluctuations in vital signs, the BVS[3] score remains low, reflecting overall physiological stability. Subsequently, during the phase of clinical deterioration, heart rate and breathing rate significantly increase while SpO$_2$ decreases compared to baseline levels. These changes lead to a high BVS[3] score crossing the threshold value of 3. In the subsequent plateau phase of the exacerbation, the BVS[3] score decreases reflecting the stabilization of deteriorated vital signs during the ongoing exacerbation. The patient later consulted a pulmonologist, leading to the initiation of antibiotic therapy, which labelled the moderate AECOPD index date. In this example, the alert triggered on the BVS[3] threshold detected the event seven days before the AECOPD index date (medical consultation).

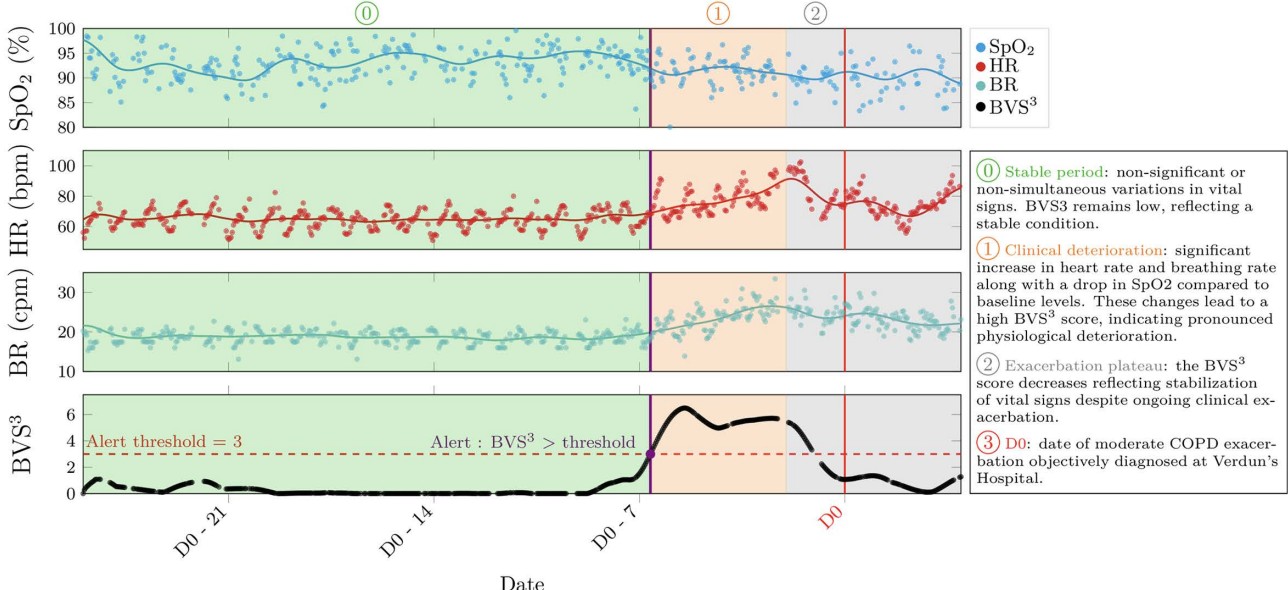

**Fig 3. Illustration of BVS³ computation.** The first three rows show the resampled SpO₂, HR, and BR, along with their Gaussian Process estimates (solid line). The final row displays the BVS³ risk score, with red dashed line indicating threshold-exceeding alerts. The patient's exacerbation diagnosis date is highlighted with a red line.

## Results

### Patient characteristics

220 patients were included in the analysis, with baseline characteristics summarized in Table 1. The cohort consisted of middle aged (average 63 years-old) predominantly male subjects (5:4 male-to-female ratio) with often multimorbid COPD, mainly classified as GOLD stage 2 and 3. At baseline, 24 patients (11%) were receiving noninvasive ventilation (NIV) and 12 patients (6%) were on long-term oxygen therapy.

### Vital signs dataset description

A total of 36,375 days of active remote patient monitoring (RPM) was analyzed, with an average RPM session duration of 173.1±21.7 days. Median compliance rate with the connected wristband was 86% (interquartile range IQR=22%). One-hour median values were available for heart rate (HR) across 511,591 instances (averaging 16±5.3 values per 24 hours), for breathing rate (BR) across 402,436 instances (16±4.8 values per 24 hours), and for oxygen saturation (SpO₂) across 343,588 instances (10.7±5.3 values per 24 hours).

### AECOPD dataset description

During the study period, a total of 58 AECOPD validated by the clinical team were documented in 51 patients at Verdun's Hospital. Four of these events were detected by the BVS³ score during the dismissed period and were therefore excluded from the analysis to minimize uncertainty about whether the detection was related to the exacerbation or to an unrelated physiological change. In 12 additional cases, the total number of vital sign measurements during the 10-day case period was below 60, which was considered insufficient to reliably evaluate the performance of the algorithm that is based on vital sign data; these AECOPD were also excluded. Consequently, a total of 42 AECOPD (7 severe and 35 moderate) occuring in 39 patients were included in the analysis. No statistical differences in baseline characteristics between patients

**Table 1. Demographic and clinical characteristics of the study population at baseline (N = 220).**

| Characteristics | Value Mean ± SD or n (%) |
|---|---|
| **N** | 220 |
| **Demographics** | |
| Age (years) | 62.65 ± 8.42 |
| Male | 122 (55%) |
| Body mass index (BMI) (kg/m²) | 27.78 ± 7.05 |
| **Spirometry** | |
| $FEV_1$ (L) | 1.62 ± 0.68 |
| pred. $FEV_1$ (% predicted) | 58.27 ± 20.34 |
| FVC (L) | 2.83 ± 0.99 |
| pred. FVC (% predicted) | 82.29 ± 20.93 |
| **Clinical Scores** | |
| mMRC dyspnea score (0–4) | 2.15 ± 1.35 |
| **Smoking Status** | |
| Current Smoker | 97 (44.1%) |
| Ex-Smoker | 114 (51.8%) |
| Non-smoker | 9 (4.1%) |
| **Respiratory Conditions** | |
| Obstructive Sleep Apnea (OSA) | 94 (42.7%) |
| GOLD stage I | 34 (15.5%) |
| GOLD stage II | 103 (46.8%) |
| GOLD stage III | 62 (28.2%) |
| GOLD stage IV | 21 (9.5%) |
| Long term oxygen therapy (LTOT) | 12 (5.5%) |
| Non-invasive ventilation (NIV) | 24 (10.9%) |
| **Comorbidities** | |
| Coronary heart disease | 41 (18.6%) |
| Diabetes | 35 (15.9%) |

Abbreviations: $FEV_1$ = Forced Expiratory Volume in 1 second; FVC = Forced Vital Capacity; NYHA = New York Heart Association classification; mMRC = Modified Medical Research Council dyspnea scale; GOLD = Global Initiative for Chronic Obstructive Lung Disease.

with excluded events (n = 12) and those with included events. The 42 corresponding case periods were compared against 2,536 control periods to assess the predictive performance of the BVS³ score.

## AECOPD prediction performance

The BVS³ score was significantly higher in time periods preceding AECOPD events compared with control periods, with median values of 3.95 for combined moderate and severe (n = 42), 3.75 for moderate (n = 35), and 4.52 for severe events (n = 7), versus 1.58 for control periods (n = 2536) (Fig 4; p < 0·001).

The BVS³ score demonstrated an AUC of 0.88 (95% CI 0.83 - 0.92) for detection of moderate and severe AECOPD events (Fig 5), with an accuracy of 84.8% and a sensitivity of 74% at 85% specificity, providing a mean anticipation time of 4.4 ± 3.1 days prior to the AECOPD event occurrence. Analysis of the 380 false-positive alerts (provided in S1 Text) showed that 16% of the corresponding control periods met at least three of the five criteria proposed by the Rome consensus (BR ≥ 24 breaths/min; HR ≥ 95 bpm; resting $SpO_2$ < 92% and/or a decrease >3%), while 61% met at least

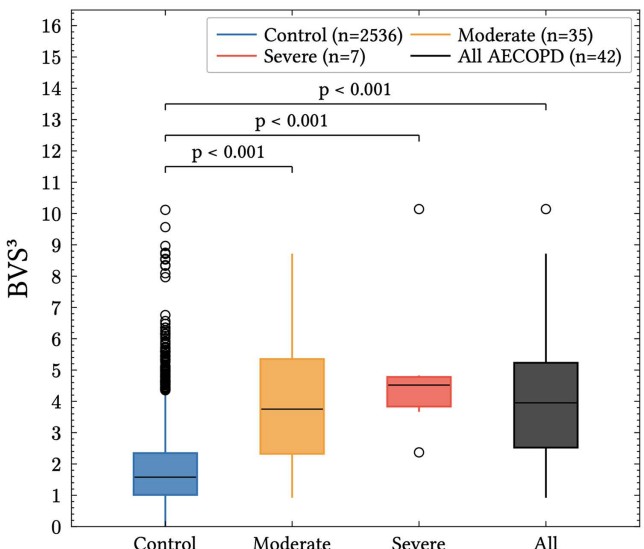

**Fig 4. Distribution of BVS³ scores in case periods preceding moderate, severe, and combined moderate/severe AECOPD events versus control periods.** Boxes represent the median and interquartile range; whiskers show the range, and markers indicate outliers. *p < 0·001, Two sampled t-test.

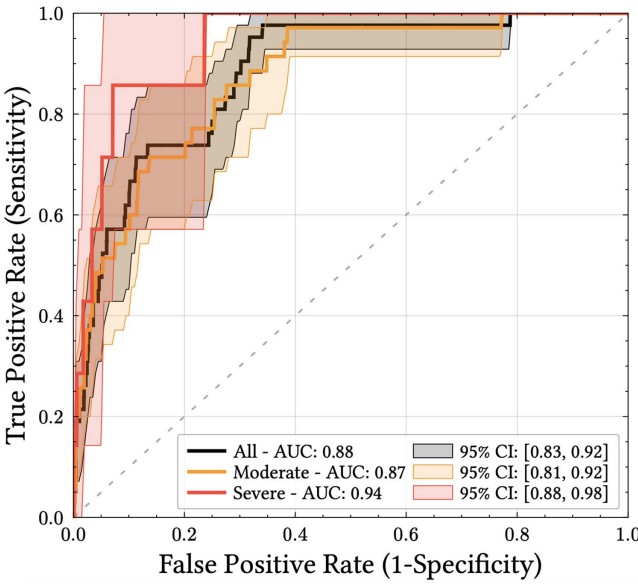

**Fig 5. ROC curves and AUC for alerts based on the BVS³ risk score (red: severe AECOPD events, orange: moderate AECOPD events, black: all events).**

two criteria and 88% met at least one, indicating they could reflect clinically meaningful physiological events that were not formally documented as exacerbations.

Performance for moderate AECOPD (n = 35) was comparable (AUC: 0.87; 95% CI 0.81 - 0.92). For the subset of severe AECOPD events (n = 7), the BVS³ score achieved an AUC of 0.94 (95% CI 0.88 - 0.98).

The performance of each specific vital sign Z-scores and every possible combination of two vital sign Z-scores was associated with a decline in performance compared to the overall BVS[3] score (Table 2).

HR and BR Z-scores showed AUC of 0.83 (95% CI 0.77 - 0.88) and 0.82 (95% CI 0.76 - 0.88) individually. BVS[3] achieved higher sensitivity, particularly at high specificity levels (Fig 6). At 90% specificity, sensitivity increased from 50% for HR or 52% for BR alone to 65% for BVS[3]. Among the 31 AECOPD events predicted by the BVS[3] score at 85% specificity, most were identified based on simultaneous high BR and HR Z-scores. However, in five cases the AECOPD early detection by the BVS[3] score relied on a high $SpO_2$ Z-score.

## Discussion

### Principal results

In this observational cohort study of 220 COPD patients with 42 exacerbations, the BVS[3] risk score derived from continuous monitoring of vital signs using a wristband demonstrated excellent predictive performance, achieving an AUC of 0.88

**Table 2. Performance of the BVS[3] algorithm and individual vital-signs Z-scores on various metrics for the early detection of AECOPD events.**

|  | AUC (95% CI) | | | Sensitivity at 85% specificity (95% CI) | |
|---|---|---|---|---|---|
|  | All AECOPD | Severe AECOPD | Moderate AECOPD | All AECOPD | Δ* vs BVS[3] |
| **BVS[3]** | **0.88 (0.83-0.92)** | **0.94 (0.88-0.98)** | **0.87 (0.81-0.92)** | **74% (60%-86%)** |  |
| **Z-BR** | 0.82 (0.76-0.88) | 0.93 (0.87-0.97) | 0.79 (0.73-0.85) | 60% (50%-78%) | -14 |
| **Z-HR** | 0.83 (0.77-0.88) | 0.87 (0.81-0.92) | 0.81 (0.75-0.86) | 62% (50%-79%) | -12 |
| **Z-SpO₂** | 0.71 (0.65-0.76) | 0.64 (0.58-0.71) | 0.71 (0.65-0.76) | 24% (12%-36%) | -50 |
| **Z-HR and Z-BR** | 0.86 (0.8-0.92) | 0.94 (0.88-0.98) | 0.84 (0.78-0.89) | 71% (58%-82%) | -3 |
| **Z-BR and Z-SpO₂** | 0.84 (0.78-0.89) | 0.94 (0.88-0.98) | 0.82 (0.76-0.88) | 72% (59%-83%) | -2 |
| **Z-HR and Z-SpO₂** | 0.82 (0.76-0.87) | 0.83 (0.77-0.88) | 0.82 (0.76-0.88) | 62% (51%-79%) | -12 |

* Absolute difference in sensitivity

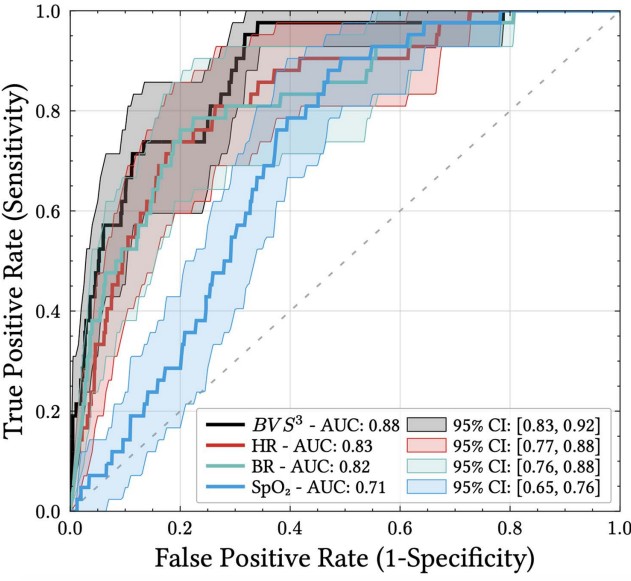

**Fig 6. ROC curves and AUC for alerts based on individual vital sign Z-scores versus the BVS[3] risk score, over all AECOPD severities.**

(95% CI 0.83 - 0.92) for moderate and severe AECOPD combined. The BVS³ score anticipated exacerbations an average of 4.4 ± 3.1 days in advance, with an overall accuracy of 84.8% and sensitivity of 74% with 85% specificity. Individual vital sign Z-scores also showed strong predictive capabilities, with breathing rate Z-score (z-BR) and heart rate Z-score (z-HR) yielding AUCs of 0.82 (95% CI 0.76 - 0.88) and 0.83 (95% CI 0.77 - 0.88), respectively.

The reliability of these results is anchored in the study's use of objective clinical data for AECOPD labeling, which avoids the potential reporting biases inherent in studies that rely on patient-administered surveys for exacerbation label-ing. Furthermore, the findings show a potential for generalizability, as they are based on a broad COPD cohort recruited from a medically underserved area, suggesting the score effectiveness can extend to diverse and less-resourced clinical settings.

### Adherence to passive remote monitoring

One of the key strengths of the monitoring approach evaluated here lies in its passive and unobtrusive data collec-tion method, which enhances its potential for scalable real-world deployment. The remote patient monitoring (RPM) system employed in this study uses a single, CE-certified Class IIa medical grade wristband to collect vital signs continuously, without requiring active input from patients. Our data demonstrated that continuous 24-hour monitoring of vital signs with a wristband was well accepted in COPD patients over the long term, with a median adherence of 86%, and enabled the collection of vital sign measurements for an average of 11–16 hours per day depending on the vital sign.

Several factors likely contributed to this high adherence. The wristband was designed for an older and frail population such as COPD patients, with a soft strap compatible with fragile skins, a screenless design to minimize patient anxiety, a long battery life and a lack of required action for data transfer or setup reducing user burden. Adherence was also rein-forced by the prospective monitoring context, as patients and families reported feeling reassured and technical issues being resolved proactively by the case manager. The 86% median adherence rate shows that these factors were efficient in mitigating patient-level barriers, thereby enabling the consistent data collection required for successful implementation of a digital intervention in COPD care which would integrate the BVS³ score.

### Predictive accuracy of the composite score

Our results validate that a patient-specific approach offers robust performance in the detection of COPD exacerbations. Alerts based on positive thresholds for HR and BR Z-scores demonstrated strong predictive capabilities (AUC 0.83 and 0.82, respectively).

Our results also confirm that a multiparameter approach combining oxygen saturation ($SpO_2$), respiratory rate (BR), and heart rate (HR) yields superior predictive performance compared to models based on one or two vital signs alone.

Although the $SpO_2$ Z-score alone showed more limited predictive performance (AUC 0.65) probably explained by sensitivity to artifacts, it provided complementary information by identifying exacerbations not detected by the HR and BR Z-scores. This contribution enhanced the overall performance of the composite BVS³ score, yielding a global AUC of 0.88 (95% CI 0.83 - 0.92) and 0.94 (95% CI 0.88 - 0.98) for severe exacerbations. These results suggest that all exacerbations are not driven by identical physiological changes, and that only the patient-specific integrated composite score can effec-tively capture the full spectrum of AECOPD events.

### Implication of the algorithmic design

The BVS³ algorithm builds upon key physiological variables derived from the Rome proposal[7], which are reflected in the 2025 GOLD recommendations [37] for classifying the severity of AECOPD. According to GOLD, this classification is based on the evaluation of accepted thresholds in clinical parameters, including three core vital signs: HR, BR, and $SpO_2$

[5]. The BVS[3] score combines these three vital signs and presents them as standardized Z-scores, allowing clinicians to interpret deviations intuitively. By relying on transparent, unsupervised methods rather than opaque black-box models, the BVS[3] score combines good predictive performance with full clinical interpretability.

Beyond interpretability, the BVS[3] score was designed for portability across monitoring systems. The Gaussian process filtering ensures robustness to data missing randomly and to various data collection rates, making BVS[3] directly transferable to other devices collecting HR, BR and $SpO_2$. Patient-specific baselines are computed automatically within the algorithm, which preserves portability across systems. This device-agnostic design enables integration into other remote monitoring platforms that centralize these vital signs.

Robustness was further strengthened by incorporating temporal filtering to mitigate effects of short-term variations on the BVS[3] score, thereby preventing the natural circadian cycle, daily-life variability and measurement artifacts from disproportionately generating false alerts — this step alone increased the global AUC from 0.84 to 0.88 and improved specificity at a 15% false-positive rate from 69% to 74%. Furthermore, the Gaussian process filter accounts for variability in baseline dispersion across patients or time periods, preventing the generation of false alerts when variability is unusually narrow and avoiding masking clinically relevant changes when variability is wide. This step further lowers the false alert rate, improving the overall usability of the score for clinicians.

Finally, alerts have been designed to be triggered only when the BVS[3] score exceeds a threshold of 3, corresponding to an expected false positive rate of 15%. This threshold balances sensitivity and specificity, ensuring a manageable workload for clinicians by keeping the false alert burden low while maintaining high detection rates.

Together, the transparency, transferability, and robustness of the algorithmic design maximize the clinical usability of the BVS[3] score and show potential to mitigate healthcare provider-level barriers to digital interventions in COPD, such as a lack of confidence in the solution or an increased workload.

### Comparison with prior work

Compared to previously published studies, the BVS[3] score demonstrates good preliminary performance and usability for real-time care: traditional long-term risk models, which rely on medical records and hospital data, have demonstrated the ability to identify patients at risk of experiencing AECOPD over the subsequent years rather than on the short term [46–50], offering limited value for real-time care. More recent methods incorporating daily symptom reports or vital signs achieved AUCs of 0.66 to 0.74 [25, 26, 28, 29], yielding lower predictive performance compared to the BVS[3] score. Some advanced multi-sensor platforms have reported higher performances (AUC up to 0.87 and 0.91) [13, 24, 31, 32], but required multiple devices, leading to increased costs, complex setup and greater patient burden due to the need for several device-specific actions. A comparative overview of prediction outcomes and horizons, data sources, predictive performance, and key limitations across prior short-term AECOPD prediction approaches is provided in Table 3, with a detailed comparison available in S1 Table.

In contrast, our single-device, passive system achieves strong performance without compromising scalability or usability necessary for widespread clinical adoption.

### Limitations

The findings should be interpreted in light of several limitations. First, the measured performance of the algorithm may be biased due to potential underreporting and misclassification of AECOPD events. Although the labeling was based on objective data sources such as hospitalization records and prescriptions, a small number of events may have been missed if patients sought treatment from their general practitioner. This potential for unlabeled AECOPD associated with abnormal biomarkers could artificially increase the number of false positives and decrease the score specificity. Furthermore, the reliance on treatment decisions and healthcare access rather than standardized symptom-based criteria from Global Initiative For chronic obstructive Lung Disease Report[37] may introduce a misclassification bias, as hospitalization can reflect care practices or social distress rather than true medical severity.

**Table 3. Comparison with prior work for short-term prediction of AECOPD.**

| Study | Prediction outcome & horizon | Data sources | Performance (AUC/ Sens@~15% FPR) | Key limitations |
|---|---|---|---|---|
| BVS[3] (this study) | Clinician-defined moderate and severe AECOPD, 10 days | Single passive wearable | Validation cohort: AUC 0.88 Sens@15% FPR = 74% | Monocentric validation; multicentric prospective validation ongoing |
| Chmiel et al., 2022 | Self-reported moderate and severe AECOPD, 3 days | Symptom questionnaire on smartphone app | AUC 0.73 Sens@15% FPR ≈ 40%* | Self-reported AECOPD without clinician/EHR confirmation; case-level data leakage risk; potential criterion contamination from symptom-based labels and predictors. |
| Glyde et al., 2023 | Self-reported moderate and severe AECOPD, 14 days | Symptom questionnaire on smartphone app | AUC 0.73 Sens@15% FPR ≈ 42%* | Self-reported AECOPD without clinician/EHR confirmation; potential criterion contamination from symptom-based labels and predictors. |
| Shah et al., 2017 | Self-reported mild, moderate and severe AECOPD, 7 days | Pulse oximeter | AUC 0.68 Sens@15% FPR ≈ 39%* | Self-reported AECOPD without clinician/EHR confirmation; likely dominance of mild exacerbations. |
| Orchard et al., 2018 | Hospitalization or OCS initiation, 1 day | Smartphone app, pulse oximeter | Severe AECOPD: AUC 0.74 Sens@15% FPR ≈ 38%* OCS initiation: AUC 0.765 Sensitivity NA | Model complexity (multi-task neural network with 153 features) exceeds available severe AECOPD (N = 55), raising overfitting and generalizability concerns. |
| Patel et al., 2021 | Self-reported mild/moderate and severe AECOPD confirmed by clinician, 14 days | Symptom questionnaire on smartphone app, spirometry, CRP | Validation cohort: AUC not computed Sens@16% FPR ≈ 98%* | Likely dominance of mild exacerbations; potential criterion contamination from symptom-based labels and predictors. |
| Wu et al., 2021 | AECOPD (unclear), 7 days | smartphone app, wearable, home air quality sensing device, environmental open data | AUC 0.985 Sens@15% FPR ≈ 95%* | Unclear AECOPD definition; model complexity (deep neural network, 45 features) exceeds available AECOPD (N = 25), raising overfitting and generalizability concerns. |
| Wu et al., 2022 | AECOPD (unclear), 7 days | smartphone app, wearable, home air quality sensing device, environmental open data | Validation cohort: AUC and ROC NA Sens@19%FPR: 69%* | Unclear AECOPD definition; incomplete reporting of AECOPD count used for training and validation, preventing assessment of overfitting risk and model robustness. |
| Jo et al., 2023 | EHR-defined moderate and severe AECOPD, 1 day | EHR, spirometry, environment | Validation cohort: AUC 0.74 Sens@61%FPR: 99%* | Unclear control selection strategy; case–control sampling not representative of real-world AECOPD prevalence, likely underestimating FPR in routine use. |
| Atzeni et al., 2025 | Self-reported mild, moderate and severe AECOPD confirmed by clinician, 1 day | Smartphone app, wearable air quality sensor, spirometer | AUC 0.87 Sens@21%FPR: 85%* | High patient burden limiting acceptability; unclear AECOPD severity distribution with likely dominance of mild AECOPD; potential criterion contamination from symptom-based labels and predictors. |

\* Sensitivity is estimated at 15% FPR using the ROC curve when available, or at the closest available operating point when the ROC curve is not available. OCS: oral corticosteroids

Second, the study's monocentric design, cohort size, and composition limit the generalizability of the findings. The high sensitivity observed for severe AECOPD (94%) at 85% specificity should be interpreted cautiously due to the small number of severe events (n = 7). While this does not affect the robustness of the main findings for moderate and severe events combined (n = 42), it may limit the precision of estimates derived from stratified analyses of the severe subgroup. Additionally, the COPD study population shows a high prevalence of comorbid obstructive sleep apnea syndrome (OSA), reflecting the original design of the eMEUSE-SANTÉ clinical trial and introducing a

potential comorbidity bias. Larger studies are needed to fully assess the generalizability and robustness of the BVS [1] score.

The algorithmic design presents its own set of limitations. The anomaly detection paradigm, which focuses on patient-specific increases in heart rate and breathing rate and decreases in $SpO_2$, may not capture atypical AECOPD patterns (for, e.g., in case of bradycardia or underlying arrythmia). Although the patient-specific baseline and temporal filtering mitigate some effects of potential confounders, including variations in physical activity or comorbidities, these factors were not explicitly included in the score computation.

Finally, the involvement of a research team at Biosency introduces the possibility of institutional bias. Although this was addressed through use of the independent academic MARS dataset (HP2 laboratory), in which all acute exacerbation dates and vital-sign time series were validated and locked before analysis, and by prespecified, quantitative development and evaluation of the score without post-hoc optimization with respect to AECOPD event dates, further validation by independent groups will be important to confirm generalizability.

## Conclusions

This study provides a pioneering validation of an unsupervised, statistical model-based score associated with a remote monitoring system for predicting AECOPD. As the first of its kind in a French cohort, it demonstrates preliminary evidence on the score's feasibility and effectiveness. Passive collection of vital signs yields strong patient compliance, and the BVS[3] risk score exhibits high accuracy and anticipation for AECOPD prediction. This score offers a pragmatic and clinically meaningful innovation in AECOPD care management by combining strong predictive power with interpretability and ease of implementation in real-world settings. It supports earlier, more personalized, and more efficient interventions that may reduce exacerbation severity and lower healthcare utilization.

These promising initial findings provide a strong rationale for larger scale trials, which would address the limitations inherent to this study for the potential OSA comorbidity bias and the monocentric design. A multicenter randomized controlled trial (NCT06523140) is planned to evaluate the generalizability of the score and to assess its impact on clinical outcomes, including quality of life and hospitalization rates. This prospective validation, alongside cost-effectiveness analyses, will be essential to fully realize the potential of this approach in improving COPD management and patient outcomes.

## Supporting information

**S1 Text. Mathematical Details of the BVS[3] score computation.**
(DOCX)

**S1 Fig. ROC curves and AUC for alerts based on the BVS[3] risk score using 7, 10 and 14-days windows.**
(DOCX)

**S2 Text. Analysis of BVS[3] false positive alerts.**
(DOCX)

**S1 Table. Detailed comparison with prior work for short-term prediction of AECOPD.**
(DOCX)

## Acknowledgments

The authors would like to thank the nurses of the Pulmonology Department at Verdun Hospital for their essential role in patient telemonitoring, and ADOR for managing the RPM equipment at patients' homes. Generative AI was used to improve the readability of the manuscript (ChatGPT OpenAI, Le Chat Mistral AI).

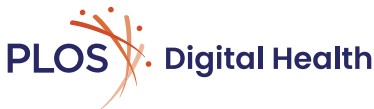

## Author contributions

**Conceptualization:** Jean-Claude Cornu, Marie Joyeux-Faure, Jean-Louis Pépin.

**Data curation:** Florian Tilquin, Sylvain Le Liepvre, Jean-Claude Cornu, Marie Joyeux-Faure.

**Formal analysis:** Florian Tilquin, Sylvain Le Liepvre.

**Funding acquisition:** Marie Pirotais, Yann Le Guillou, Jean-Claude Cornu, Marie Joyeux-Faure, Jean-Louis Pépin.

**Investigation:** Jean-Claude Cornu.

**Methodology:** Jean-Claude Cornu, Marie Joyeux-Faure, Jean-Louis Pépin.

**Project administration:** Marie Pirotais, Yann Le Guillou.

**Supervision:** Marie Pirotais, Yann Le Guillou.

**Visualization:** Florian Tilquin.

**Writing – original draft:** Florian Tilquin, Sylvain Le Liepvre, Soumaya Balbolia.

**Writing – review & editing:** Florian Tilquin, Sylvain Le Liepvre, Soumaya Balbolia, Marie Pirotais, Yann Le Guillou, Nicolas Roche, Gerard Criner, Marie Joyeux-Faure, Jean-Louis Pépin.

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
