## [Decision Letter · Decision Letter 0]

22 Jan 2026

PDIG-D-25-01009Prediction of COPD exacerbations using a machine learning score based on wearable vital sign monitoringPLOS Digital Health Dear Dr. Le Liepvre, Thank you for submitting your manuscript to PLOS Digital Health. After careful consideration, we feel that it has merit but does not fully meet PLOS Digital Health's publication criteria as it currently stands. Therefore, we invite you to submit a revised version of the manuscript that addresses the points raised during the review process. Please submit your revised manuscript by Mar 23 2026 11:59PM. If you will need more time than this to complete your revisions, please reply to this message or contact the journal office at digitalhealth@plos.org.  Please include the following items when submitting your revised manuscript:* A letter that responds to each point raised by the editor and reviewer(s). You should upload this letter as a separate file labeled 'Response to Reviewers'. This file does not need to include responses to any formatting updates and technical items listed in the 'Journal Requirements' section below.* A marked-up copy of your manuscript that highlights changes made to the original version. You should upload this as a separate file labeled 'Revised Manuscript with Track Changes'.* An unmarked version of your revised paper without tracked changes. You should upload this as a separate file labeled 'Manuscript'. If you would like to make changes to your financial disclosure, competing interests statement, or data availability statement, please make these updates within the submission form at the time of resubmission. Guidelines for resubmitting your figure files are available below the reviewer comments at the end of this letter. We look forward to receiving your revised manuscript. Kind regards, Stanley J. Szefler, MDGuest EditorPLOS Digital Health Leo Anthony CeliEditor-in-ChiefPLOS Digital Healthorcid.org/0000-0001-6712-6626 **Journal Requirements:** If the reviewer comments include a recommendation to cite specific previously published works, please review and evaluate these publications to determine whether they are relevant and should be cited. There is no requirement to cite these works unless the editor has indicated otherwise.  **Additional Editor Comments (if provided):** Thank you for submitting your manuscript for potential publication to PLOS Digital Health. You will see attached some excellent comments from three reviewers familiar with the process you are describing and the role in disease management. Some of the comments may be easy to answer and some may require re-analysis of your system. The strengths of your report are in the innovation and potential impact on management of chronic obstructive pulmonary disease. Some weaknesses described relate to the small event count and analytical approach, especially the potential for false alarms. You also should be clear on the potential for conflicts of interest since four authors are Biosensory employees/founders and three clinical authors serve on Biosensory's scientific advisory board. We look forward to reviewing your response to the comments and your revised manuscript proposal.**Reviewers' Comments:** Reviewer's Responses to Questions

**Comments to the Author**

1. Does this manuscript meet PLOS Digital Health’s publication criteria? Is the manuscript technically sound, and do the data support the conclusions? The manuscript must describe methodologically and ethically rigorous research with conclusions that are appropriately drawn based on the data presented.

Reviewer #1: Yes

Reviewer #2: Yes

Reviewer #3: Yes

2. Has the statistical analysis been performed appropriately and rigorously?

Reviewer #1: Yes

Reviewer #2: No

Reviewer #3: N/A

3. Have the authors made all data underlying the findings in their manuscript fully available (please refer to the Data Availability Statement at the start of the manuscript PDF file)?

Reviewer #1: Yes

Reviewer #2: No

Reviewer #3: Yes

4. Is the manuscript presented in an intelligible fashion and written in standard English?

Reviewer #1: Yes

Reviewer #2: Yes

Reviewer #3: Yes

5. Review Comments to the Author

Reviewer #1: This paper proposes an approach to detect early detection of acute exacerbations of chronic obstructive pulmonary disease using remote patient monitoring. The developed method has been evaluated using a collected dataset to validate its performance. The study is important; however, some details of the study could be further elaborated.

Line 17: Provide the full name of BVS3, as this is the first time this abbreviation appears.

Line 40: "advanced multi-sensor systems", please name examples of such systems.

Line 82: The BVS3 score was already developed in [21] by the authors? If so, then it cannot claim the development of this score as part of the contributions of this paper. Please clarify.

Line 109: "It is an unsupervised algorithm that combines predefined mathematical formulas with machine learning". I didn't see any machine learning component (e.g., classification model) in the calculation of BBS3 scores. It is fully determined by the statistics, so it is a statistical approach. Therefore, it might not be accurate to call it a machine learning approach.

Line 169: HR, BR, and SpO₂ are used in the proposed approach. Besides the practical consideration of continuous measurement, are there any other justifications (from the machine learning perspective, such as feature importance) for the choice of these signals?

Line 183: "Gaussian Process filtering was applied to denoise, interpolate missing values." The hyperparameter of the filtering should be documented, such as the sliding window size.

Line 210: It would be good to present two figures to visually compare the BVS3 score in two scenarios (normal vs. alert).

Line 312: "Robustness was further strengthened by incorporating temporal filtering to mitigate effects of short-term variations." It would be good to show how many false alerts were reduced.

Line 325: It is very important to compare with the previous study. Please clarify the details, such as the experimental setup. Also, it would be good to use a table to compare the performance.

Reviewer #2: Thank you for submitting this manuscript examining the BVS3 score for predicting COPD exacerbations using wearable vital sign monitoring. The work addresses a clinically meaningful problem and demonstrates promising feasibility data. However, several issues require attention.

Major issues:

1. The characterization of BVS3 as a "machine learning score" is imprecise and potentially misleading. The algorithm employs predefined mathematical formulas with Gaussian process signal processing for data interpolation, not supervised learning trained on outcomes. Please revise the title, abstract, and throughout to accurately describe the methodology (e.g., "algorithm-based" or "signal-processing-based" risk score).

2. The paper has 42 AECOPD events with "no missing remote monitoring data." Please clarify: How many total AECOPD events occurred during the study period? What proportion were excluded due to missing data, and did excluded patients differ systematically from included patients? This information is essential for assessing generalizability. Maybe I missed it, but I re-read again trying to find, even if it said 86% followed the protocol.

3. Sample size concerns - this is different from missing data above. The severe AECOPD subgroup (n=7) is too small for reliable inference. While acknowledging this limitation is appreciated, consider removing or substantially downweighting the severe-specific analyses, or present these findings explicitly as exploratory with wider confidence intervals.

4. The manuscript would benefit from direct comparison with existing validated prediction tools. Could you apply the ACCEPT tool or similar established models to your cohort to provide benchmarking context?

5. False-positive analysis: Please provide more detailed characterization of false positive alerts. What clinical events or circumstances triggered false alarms? This information is critical for understanding clinical workflow implications.

Minor issues:

6. Table 2 formatting: The "delta vs BVS3" column shows percentage point differences in sensitivity but could be misinterpreted. Please clarify these are absolute differences, not relative changes.

7. Please specify the bootstrap resampling parameters (number of iterations, resampling strategy) used for confidence interval estimation.

8. Threshold selection - The threshold of 3 was chosen based on "preliminary data" for 85% specificity. Please provide more detail on this calibration dataset and whether any overfitting concerns exist.

9. While COI disclosures are provided, please add a statement in the Discussion explicitly acknowledging how industry involvement was managed to ensure scientific integrity.

Reviewer #3: Comments on COPD paper D-25-01009

This is overall an interesting and well-constructed paper on an important topic appropriate to PLOS Digital Health. I provide comments below. Some are not entirely trivial. However, I expect that they can all be addressed without undue difficulty by the authors. Also, AI believe that carefully addressing these questions will substantially boost the impact of the paper. Thanks for this opportunity to read about an interesting and useful study.

(line number : comment. The most important comments are preceded by **.)

17 - 18 : Perhaps clarify this sentence: "this study has two components. It introduces ... , and also retrospectively..." Or something like this. The sentence is not clear as-is.

** 25 - 26 : (this comment comes up multiple times) What is the connection/relevance of these results to clinical use cases?

75 : Perhaps you can provide a more general context, with citations, of wearables for clinical use cases (Sibel, Oura, etc). Then narrow down to wearables for COPD.

** 92 - 95 : (this comment comes up multiple times) Please discuss what clinicians want - any ML or automated intervention should be placed in this context. For example, if clinicians want automatic warnings of potential AECOPD, what sensitivity do they need? Do they need warnings for mild exacerbations (mentioned in Rome, not covered in this paper)? What is their tolerance for "alert fatigue" (in this paper, 85% specificity would give, on this cohort, 400 false alarms for 20 - 30 true alarms; is this acceptable?)

** 101, 125 : (this comment comes up multiple times) You note that the wearable device recorded physical activity. Why was this information not used? This is relevant because physical activity affects HR, BR, and maybe SpO2, and the Rome protocols specify resting rates of HR and BR. Also, does sleep represent a distinct type of activity with its own AECOPD diagnosis rubric? Sleep can be likely be inferred from the device readings, bolstered by a bayesian prior that it occurs at night.

103 : What is the impact of not having controls (ie healthy people)? Normally, controls are a valuable resource to set baseline or noise levels. This paper uses time periods distant from a clinically-recorded AECOPD episode. Is this an adequate set of controls? (I don't know, but the question certainly comes up for the reader)

109 - 110 : Is "machine learning" correct here, or are these just signal processing methods. I think it's the latter, which is actually preferable (since then no training data is required) but it should be clarified.

** 151 : The Rome protocol used 14 days, implying that a 13-day prior warning is not dissociated from the AECOPD event. So 14 days is the implicit default choice. The question arises, why was this not used in this study, and was 10 days chosen post-hoc since it gave the sharpest results?

156, 160 : Is "30" a typo? Fig 2 indicates that this should be "20". Also, based on line 160, the period 20 - 30 days prior is not accounted for. The "dismissed period" choice makes good sense.

** 164 : what is the clinical basis for 85% specificity? This results in a 20:1 ratio of false alarms to true alarms in this cohort.

** 169 - 179 : The Rome protocol specifies "resting" vitals. However, the mean and std dev appear to be over all times. Would it be more appropriate to (a) use a Mahalanobis distance based on the distribution of resting vital sign values (mean and std dev will be lower than for all data points); and (b) restrict anomaly detection to just resting periods (based on activity detection data), to conform to the Rome protocol and avoid false alarms due to physical activity? This is a nontrivial issue, because activity could generate many false alarms. That is, your specificity might get much better at no cost to sensitivity if you restrict to resting periods.

** 168 ff : Can the BSV distinguish moderate vs severe AECOPD, or is it purely binary? What about mild cases? What are the clinical implications of this?

183 : Were there any rejection criteria, eg for values that are simply unrealistic (eg zero motion, HR < 40, temperature < 34, etc)?

** 189 : "with each contributing equally". This is very surprising, because there is no a priori reason (that I know of) that these symptoms should have equal salience. Is there a reason for equal weights vs calibrating individual weights on a test set?

192 - 193 : Perhaps take a moment to walk the reader through this equation, so that the reader is not bogged down unnecessarily.

** 193 : I do not understand the "2" in eqn 3. It looks like you are capturing the probability of both tails, but it is unclear why this is desirable. The adverse conditions are in one direction only (ie elevated HR, BR, 100 - SpO2). So would a one-sided tail would be appropriate? On that topic, would a one-sided std dev, calculated only on points greater than the mean, be a more accurate parameter?

** 197 : "threshold value of 3": on what data was this determined? It must be different data from this cohort, or else the cohort needs to have a k-fold split (stratified by patient) to make the threshold selection independent of the results. Also (as noted above) what is the clinical justification for 85% specificity?

213 : are there any patient characteristics that would affect PPG readings, eg skin color?

213 (and "limitations" also): How does the overall sickness of this cohort relate to the expected cohorts in the envisioned use cases? For implications, see for example Kendall "Whom tuberculosis tests detect and why it matters" 2025.

251 : As noted above, 85% specificity gives a very high number of false alarms. For a way to assess ROC curves in this context, see Delahunt "Metrics to guide development of ml algorithms for malaria diagnosis" 2024.

302 : "aligned with the 2025 GOLD recommendations [24]": these come directly from the Rome protocols [5]. Perhaps note this for clarity.

310 : Perhaps note that each patient needs to have a baseline distribution for each vital sign to use the method.

** 325 : "high performance and usability for real-time care": this is overly optimistic, unless the results are adequate for concrete clinical use cases. Eg it must have sufficient sensitivity and few enough false alarms; the cohort of this study is sufficiently representative of expected cohorts in known use cases.

343 - 344: "the very high specificity": this is baffling. It does not agree with the plots, which show quite low specificity at moderate sensitivities.

6. PLOS authors have the option to publish the peer review history of their article (what does this mean?). If published, this will include your full peer review and any attached files.

**Do you want your identity to be public for this peer review?** For information about this choice, including consent withdrawal, please see our Privacy Policy.

Reviewer #1: No

Reviewer #2: **Yes:** SAPTARSHI PURKAYASTHA

Reviewer #3: No

  **Figure resubmission:**  While revising your submission, we strongly recommend that you use PLOS’s NAAS tool (https://ngplosjournals.pagemajik.ai/artanalysis) to test your figure files. NAAS can convert your figure files to the TIFF file type and meet basic requirements (such as print size, resolution), or provide you with a report on issues that do not meet our requirements and that NAAS cannot fix. 

After uploading your figures to PLOS’s NAAS tool - https://ngplosjournals.pagemajik.ai/artanalysis, NAAS will process the files provided and display the results in the "Uploaded Files" section of the page as the processing is complete. If the uploaded figures meet our requirements (or NAAS is able to fix the files to meet our requirements), the figure will be marked as "fixed" above. If NAAS is unable to fix the files, a red "failed" label will appear above. When NAAS has confirmed that the figure files meet our requirements, please download the file via the download option, and include these NAAS processed figure files when submitting your revised manuscript. **Reproducibility:** To enhance the reproducibility of your results, we recommend that authors of applicable studies deposit laboratory protocols in protocols.io, where a protocol can be assigned its own identifier (DOI) such that it can be cited independently in the future. Additionally, PLOS ONE offers an option to publish peer-reviewed clinical study protocols. Read more information on sharing protocols at https://plos.org/protocols?utm_medium=editorial-email&utm_source=authorletters&utm_campaign=protocols

---

## [Decision Letter · Decision Letter 1]

16 Apr 2026

Short-term prediction of COPD exacerbations based on wearable vital sign monitoring

PDIG-D-25-01009R1

Dear Dr Le Liepvre,

We are pleased to inform you that your manuscript 'Short-term prediction of COPD exacerbations based on wearable vital sign monitoring' has been provisionally accepted for publication in PLOS Digital Health.

Before your manuscript can be formally accepted you will need to complete some formatting changes, which you will receive in a follow-up email from a member of our team

Best regards,

Stanley J. Szefler, MD

Guest Editor

PLOS Digital Health

**Additional Editor Comments (if provided):**

Thank you for your thorough response to the comments from the three reviewers. Although one reviewer, Reviewer #2 declined the opportunity to review your response, I reviewed your response. I believe your response is thorough and adequately addresses the concerns previously expressed by Reviewer #2. Reviewer #3 identified two typographical errors that could be easily corrected before final acceptance.

**Reviewer Comments (if any, and for reference):**

Reviewer's Responses to Questions

**Comments to the Author**

1. If the authors have adequately addressed your comments raised in a previous round of review and you feel that this manuscript is now acceptable for publication, you may indicate that here to bypass the “Comments to the Author” section, enter your conflict of interest statement in the “Confidential to Editor” section, and submit your "Accept" recommendation.

Reviewer #1: All comments have been addressed

Reviewer #3: (No Response)

2. Does this manuscript meet PLOS Digital Health’s publication criteria? Is the manuscript technically sound, and do the data support the conclusions? The manuscript must describe methodologically and ethically rigorous research with conclusions that are appropriately drawn based on the data presented.

Reviewer #1: Yes

Reviewer #3: Yes

3. Has the statistical analysis been performed appropriately and rigorously?

Reviewer #1: Yes

Reviewer #3: Yes

4. Have the authors made all data underlying the findings in their manuscript fully available (please refer to the Data Availability Statement at the start of the manuscript PDF file)?

Reviewer #1: Yes

Reviewer #3: Yes

5. Is the manuscript presented in an intelligible fashion and written in standard English?

Reviewer #1: Yes

Reviewer #3: Yes

6. Review Comments to the Author

Reviewer #1: The revision is fine.

Reviewer #3: Thank you for your thorough and thoughtful responses to all the reviewers. I believe the paper's impact is much improved as a result of your clarifications and additions.

Two typos:

line 20: "evaluates" -> "evaluate"

line 194: "D-20 to D0" -> "D-20 to D-10"

7. PLOS authors have the option to publish the peer review history of their article (what does this mean?). If published, this will include your full peer review and any attached files.

**Do you want your identity to be public for this peer review?** For information about this choice, including consent withdrawal, please see our Privacy Policy.

Reviewer #1: No

Reviewer #3: No
